# Statistical Considerations for Trials in Adjuvant Treatment of Colorectal Cancer

**DOI:** 10.3390/cancers12113442

**Published:** 2020-11-19

**Authors:** Everardo Delforge Saad, Marc Buyse

**Affiliations:** 1International Drug Development Institute, 1340 Louvain-la-Neuve, Belgium; everardo.saad@iddi.com; 2Dendrix Research, Sao Paulo 04534-000, Brazil; 3Interuniversity Institute for Biostatistics and Statistical Bioinformatics (I-BioStat), Hasselt University, 3590 Diepenbeek, Belgium

**Keywords:** adaptive trials, adjuvant therapy, biomarkers, clinical trials, colorectal neoplasms, minimization, neoadjuvant therapy, overall survival, patient-centricity, randomized clinical trials, surrogate endpoints

## Abstract

**Simple Summary:**

Improvements in the treatment of patients with cancer largely depend on clinical trials comparing a new intervention with a previous standard of care. This is not different when it comes to improving adjuvant (postoperative) and neoadjuvant (preoperative) treatment for colorectal cancer. In clinical trials, there is usually a close collaboration between physicians and statisticians, and better and more reliable results can be obtained when such trials are designed, conducted and analyzed with attention to some important methodological issues. In this article, we provide a general review of useful methodological and statistical ideas that may help investigators and clinicians in their attempt to improve treatment results in colorectal cancer.

**Abstract:**

The design of the best possible clinical trials of adjuvant interventions in colorectal cancer will entail the use of both time-tested and novel methods that allow efficient, reliable and patient-relevant therapeutic development. The ultimate goal of this endeavor is to safely and expeditiously bring to clinical practice novel interventions that impact patient lives. In this paper, we discuss statistical aspects and provide suggestions to optimize trial design, data collection, study implementation, and the use of predictive biomarkers and endpoints in phase 3 trials of systemic adjuvant therapy. We also discuss the issues of collaboration and patient centricity, expecting that several novel agents with activity in the (neo)adjuvant therapy of colon and rectal cancers will become available in the near future.

## 1. Introduction

After progressive increase over some decades, research output on adjuvant therapy for colorectal cancer seems to have stalled in the last few years. As suggested by a simple search for published articles (Figure 1), the number of research publications and potentially research output on adjuvant therapy for colorectal cancer has decreased since 2016. Moreover, apart from refinements, the adjuvant therapy of colorectal cancer has not changed considerably in recent years. However, the ongoing revolution of targeted therapy, immunotherapy and precision oncology is reason to expect that several novel agents and predictive biomarkers will become available and undergo testing in this setting in the near future. Likewise, lifestyle modification and interventions other than antineoplastic therapy appear to be on the rise, and their use will require solid evidence from randomized trials. The statistical design of the best possible clinical trials of these agents and interventions will entail the use of both time-tested and novel methods that allow efficient, reliable and patient-relevant therapeutic development. The ultimate goal of this endeavor is to safely and expeditiously bring novel agents and interventions that impact patient lives to clinical practice. In an attempt to offer a succinct guide to investigators planning a trial on systemic adjuvant therapy for adenocarcinomas of the colon and rectum, we discuss statistical aspects and provide suggestions to optimize trial design, data collection and the use of biomarkers and endpoints in trials. The focus of our discussion will be on phase 3 trials for antineoplastic drugs.

## 2. Optimizing Aspects Related to Trial Design 

### 2.1. Overall Objectives of the Trial

The prominence of phase 3 trials in drug development and as the evidence base for clinical practice stems from their ability to infer causal effects of the randomized treatments on the outcomes of interest. Given their explicit objectives and statistical-testing framework, these trials provide the ideal setting for comparing two or more competing interventions in a manner that accounts for confounding factors, such as baseline patient characteristics and ancillary interventions that can be controlled to some extent. A property of experimental research in general, and phase 3 trials in particular, is that internal validity (i.e., the reliability of results) and external validity (i.e., their generalizability) tend to move in opposite directions as investigators attempt to control experimental features such as the study population, the intervention, supportive and post-trial treatment and the assessment of outcomes. As a result, two different polar attitudes toward phase 3 trials are generally recognized: one that prioritizes internal validity (the “explanatory attitude”), and one that places more emphasis on the generalizability of results (the “pragmatic attitude”) [1]. Pivotal trials of novel agents, often sponsored by pharmaceutical companies, are typical examples of the explanatory attitude [2], whereas investigator-led and pragmatic trials, often conducted by academic groups or governmental agencies, often conform to the pragmatic attitude [3]. In essence, the former type of trials prioritizes the assessment of “efficacy”, whereas the latter prioritizes “effectiveness” in real life. Table 1 displays these and other features of explanatory and pragmatic trials considering polar situations. In practice, many trials display elements of both attitudes, and tools are available to determine the relative position of a trial in the explanatory–pragmatic continuum [4]. If only the polar cases are considered, foremost among the differences in “attitudes” between explanatory and pragmatic trials respectively, are the primary purpose (regulatory approval vs. improving clinical practice), the patient profile (more vs. less selected on the basis of eligibility), the control of variables (more vs. less) and data collection (more vs. less). 

When optimizing trial design, consideration should be given to the relative position in that continuum. Industry-sponsored clinical trials, essential for drug development, need to fulfil commercial interests, not always addressing the gaps that exist in clinical practice. Moreover, pivotal trials have to fulfil regulatory requirements put in place to ensure a new drug is sufficiently safe and active. However, there is often a need to generate additional, post-approval evidence on novel drugs, as this evidence is required for clinical practice and understanding of the “real-life” performance of the treatment, for example, in populations or settings not covered by the initial approval. In some cases, many questions of clinical interest remain unanswered at the time of approval, such as the duration of therapy, dose modifications that may lead to a better benefit/risk ratio and combinations with existing interventions. Even though it may be tempting to answer these questions through observational studies, randomized trials remain the only reliable design to infer causality between treatment and outcomes [5]. For all these reasons, investigator-led clinical trials are paramount in the attempt to improve the therapeutic arsenal in oncology. When planning these trials, investigators should borrow several of the time-tested and scientifically sound approaches and procedures used in pivotal trials, but there is ample room for gaining efficiency and containing costs if a rational attitude is adopted toward practices related to data collection and management, as discussed in more detail below and elsewhere [6]. 

Another consideration with regard to phase 3 trials is their main statistical objective. Whereas most phase 3 trials aim to demonstrate the superiority of the experimental therapy in comparison with control, non-inferiority phase 3 trials may also assess whether a more convenient, less toxic, or more affordable intervention displays similar efficacy to an existing standard of care [2,3]. A final general consideration pertains to the various types of analyses that can be done by varying the analysis population, the definition of the endpoint in question and the methods used to deal with intercurrent events and missing data. These various aspects have implications on the “treatment effect” ultimately determined in a given analysis, and they can be jointly considered under the recently proposed framework of “estimands”, which provides an attempt to prespecify and make explicit these various choices [7]. Traditionally, an intent-to-treat (ITT) analysis that includes all randomized patients, regardless of the actual treatment delivered, is the most appropriate, mainly because randomization creates comparable groups in terms of known and unknown prognostic factors. If anything, lack of exposure to treatment or outcome assessment is expected to bias results toward the null hypothesis, which is considered as appropriately conservative in superiority trials. Exclusion of patients from analysis, for example when per-protocol analyses are done, may introduce bias. Special considerations may be required in non-inferiority trials, a setting in which there is still debate about whether ITT or per-protocol analyses should take precedence [8]. Although the ITT analysis remains the most important and the one able to exploit the advantages of randomization to the greatest extent possible, there are increasing concerns about whether estimating an effect in accordance with the ITT principle always represents the treatment effect of greatest relevance in all situations [7,9]. In the current paper, we will not address the limitations of the ITT analysis, and the reader is referred to regulatory and review pieces on this subject [7,9,10].

### 2.2. Trial “Anatomy”

By “anatomy” here we mean the overall format of the trial in terms of number of arms, the unit of randomization, the existence of a true control and the possibility for a cross-over. The vast majority of phase 3 trials have one experimental and one true control arm, by which we mean an existing intervention, placebo, observation, or best-supportive care. In the adjuvant setting, most trials have two arms, an ideal framework in terms of efficiency and ease of interpretability [11,12]. However, two or more experimental arms may be tested [13], as these multi-arm trials can save overall resources by having a common control arm, despite enrolling a larger number of patients than two-arm trials [14]. The statistical design for such multi-arm trials may require special attention with regard, for example, to adjustment for multiplicity. With further statistical refinements and prespecified adaptations, the so-called multi-arm, multi-stage trials, allow simultaneous assessment of various novel treatments against a single control arm, with discontinuation of arms that do not show sufficient promise [15]. Additional modifications to the general anatomy of phase 3 trials are not frequently used in the adjuvant setting, and they include factorial trials, cluster-randomized trials and trials with an inbuilt two-way cross-over, the latter being rare even in the advanced disease setting [16].

### 2.3. Incorporating Biomarkers in Trial Design

Biomarkers can be of several types depending on their function, but the relevant ones in the present discussion are prognostic and predictive biomarkers. Prognostic biomarkers identify the likelihood of a clinical event, regardless of treatment type, whereas predictive biomarkers identify individuals more likely to benefit from a given treatment. Surrogate biomarkers are discussed in Section 3.2.

Cancer research is undergoing unprecedented changes due mostly to increasingly sophisticated knowledge about the biological features of tumors, the availability of a growing number of highly effective therapies and an evolving regulatory environment. As a result of these changes, precision oncology can fulfil its promise of taking advantage of actionable targets on the basis of predictive biomarkers. This leads to a general anticipation that precision oncology can offer vastly superior results, as compared with those observed in the chemotherapy era. However, the current concept of precision oncology is likely oversimplified, because tumor heterogeneity is so complex and dynamic that no two patients have the same disease from a molecular viewpoint, and a single patient has tumor lesions with different genetic landscapes both in space and in time [17]. In reality, given the inherent limitations in knowledge about tumor biology and the combined use of agents with different targets, only one or a few molecular alterations are taken into account for treatment decisions. Arguably, this will still be the case in adjuvant therapy for years to come, especially in the setting of the large randomized trials required to demonstrate gains in long-term outcomes. Even if precision oncology eventually becomes the norm in the adjuvant therapy of colorectal cancer, we and others have proposed that randomized trials will still be needed [18]. One of the reasons for this proposal is that the prognostic role of a large number of known and unknown patient and tumor features, including biomarkers, may lead to selection bias affecting any attempt to infer causality between treatment and outcomes. Even though known confounding variables may be taken into account through design and analysis (e.g., through stratification and multivariable analysis), unknown confounders can only be dealt with through randomization. Of note, historical comparisons—such as those usually entailed in real-world studies—are often threatened by the additional confounding effect of stage migration and temporal trends in supportive care and other factors. Importantly, the predictive role of biomarkers cannot be ascertained in a definitive fashion without randomization to a control arm. The very concept of a predictive biomarker requires demonstration that a given treatment is superior to control in a manner that varies according to the level of expression of the biomarker. Therefore, the lack of a control arm does not allow differentiation between the prognostic and the predictive role of a biomarker. 

We believe the main challenge in designing clinical trials for precision oncology in the adjuvant setting is to ensure the predictive biomarker is already at a stage of sufficient analytical and clinical validation. In many cases, testing of predictive biomarkers in the adjuvant setting is performed in a retrospective fashion, through subgroup analysis [19]. Such retrospective analyses are useful for hypothesis generation, but they are often threatened by increased risks of false-positive results, and the best way to confirm the predictive impact of a biomarker (i.e., whether the value taken by this biomarker predicts for a larger or smaller treatment effect) is to conduct a prospective “interaction” or “biomarker-stratified” design, in which patients are first stratified for the value of the biomarker, and then randomized to the available treatment options (Figure 2). Other biomarker-based designs are available, but we believe they are less useful for phase 3 trials on adjuvant therapy [20].

### 2.4. Aspects Related to Randomization

Randomization, the crown jewel of clinical research, is able to balance the groups being compared if the sample size is large enough for known and, more importantly, for unknown prognostic factors, thus avoiding selection bias. So-called accidental bias may still remain, whereby groups will differ by important prognostic characteristics only by chance; this happened, for example, in the Pan European Trial Adjuvant Colon Cancer (PETACC)-3 trial, in which proper randomization resulted in a statistically significant excess of patients with T4 tumors in one arm even in a trial with nearly two thousand patients [21]. It is worth remembering that a properly randomized trial must display two essential features: the unpredictability of the allocation sequence and proper concealment of such sequence. Of the several aspects of randomization deserving attention, here, we only cover the benefit of some methods of adaptive randomization and the role of unequal randomization, leaving aside the equally important issues of blinding, concealment and the required infrastructure around these issues. 

Although there is some disagreement among statisticians, simple randomization—in which no permuted blocks or stratification factors are used—is generally recommended only for very large trials. Permuted blocks are sequences of random assignments that optimize balanced numbers of patients across trial arms at any given point. Even in the adjuvant setting, oncology trials are typically not large enough for simple randomization, and the use of permuted blocks and stratification are essential features unless minimization is used, as discussed below. All these are features of “fixed randomization”, in which the allocation ratio is fixed a priori. Another form of randomization is “adaptive randomization”, a group of techniques that take into account ongoing trial features before specifying the probability of allocation to each group. This may be done in at least two manners, one that only takes into account patient baseline characteristics of interest—i.e., in an agnostic manner regarding the treatment effect—and one that takes the treatment effect into account. We would strongly advocate the former, and strongly discourage the latter [22]. The former is best represented by minimization, a dynamic allocation method for which there is no predefined sequence, and each patient is randomized in a way that minimizes the imbalance in predefined prognostic features across treatment arms [23]. Consider the example of a new patient randomized using minimization in a trial that compares stopping vs. continuing adjuvant chemotherapy after 3 months, as in the International Duration Evaluation of Adjuvant Therapy (IDEA) Collaboration [3]. The minimization algorithm tries to reduce the overall imbalance in terms of four baseline factors: the type of chemotherapy (FOLFOX vs. CAPOX), T stage (T1–3 vs. T4), N stage (N1 vs. N2) and center. Table 2 shows the number of patients already randomized to each arm (“Stop” or “Continue”) who have the same features as the new patient to be randomized in terms of chemotherapy (FOLFOX), T stage (T4), N stage (N2) and center (Jules Bordet Institute). In this numerical example, the new patient will be allocated to “Continue” because the column total corresponding to the “Continue” policy is smaller than the column total corresponding to the “Stop” policy. Hence, the last row in Table 2 (“Total”) shows what is called “overall imbalance”, which is reduced if the new patient is allocated to “Continue”, because this action would bring “100” and “96” closer to each other. It should be noted that minimization can be implemented in a manner that ensures unpredictability of each allocation if it is made in a stochastic rather than deterministic fashion. In other words, the next patient in the above example would not be automatically allocated to “Continue”, but rather randomized to “Stop” vs. “Continue” with a ratio (e.g., 1:4) favoring “Continue”.

Response-adaptive (or outcome-adaptive) randomization is very different in nature from such covariate-adaptive randomization, because it uses the treatment responses of all previously randomized patients to allocate with higher probability a new patient to the treatment arm that has the better outcome so far. Although this approach may appear intuitively appealing, it raises serious statistical as well as ethical concerns that make its use unadvisable [24,25,26]. Moreover, even if these concerns could be ignored, the gains to be expected from outcome-adaptive randomization are generally so small as to make the complexity of this approach unwarranted [27].

Although randomization is more often equal (i.e., a 1:1 ratio), because it is intuitively in accordance with the principle of equipoise and because it optimizes the relationship between power and sample size (all else being equal), unequal fixed randomization is very useful in settings for which there is already substantial information about the control arm, and thus having more patients (e.g., twice as many) in the experimental arm may maximize the use of patient information about the novel treatment at the expense of a small increase in sample size (of the order of 10–15% for 2:1 ratios). Although phase 3 trials in advanced cancer often use a 2:1 randomization, we are not aware of this practice in the adjuvant setting in colorectal cancer. However, we see no reason why it could not be used, even though regimens tested in the adjuvant setting are usually well known from the previous experience in advanced disease. Concerns have been raised about using the greater chance of receiving the novel therapy as an argument to convince patients to participate in a clinical trial. However, we submit that unequal randomization, which does not violate the principle of clinical equipoise, cannot be justified on the basis of ethics, but rather and simply on the wish to generate more data on a novel therapy than on a well-known control. 

### 2.5. Interim Analyses and Adaptive Designs

Since phase 3 trials on adjuvant therapy are relatively large and take years to be completed, it may be necessary to plan for frequent interim analyses, possibly using surrogate markers of efficacy, in order to stop the study as early as possible, and minimize the number of patients exposed to an inferior treatment if interim data confirm the superiority of the alternative treatment. One of the challenges in trials of adjuvant therapy is the long time needed for a large enough number of events to be observed, and for a meaningful interim analysis to take place, for instance with half the number of events required for the final analysis. If the patient accrual rate in the trial is fast, as is desirable, it is likely that the trial will be fully accrued before half the number of events are observed. In such a scenario, the role of interim analyses is limited, as is the room for adapting the trial based on interim results, for instance by increasing its sample size, or by enriching the trial in some patient subsets with more promising benefits from the experimental therapy. 

## 3. Optimizing Aspects Related to Endpoints 

### 3.1. Definition of the Endpoint

Historically, overall survival (OS) has been considered the most adequate primary endpoint for trials of adjuvant therapy in different settings, including colorectal cancer. However, the increasing effectiveness of such therapy and number of salvage-treatment options has led to the need for increasingly longer follow-up and sample sizes in order to demonstrate significant gains in OS from adjuvant therapy [28]. As a consequence, other time-to-event endpoints have progressively replaced OS as primary endpoint in this setting. 

A series of attempts have been made to harmonize definitions of time-to-event endpoints for adjuvant therapy in different cancer types [28,29,30]. In colon cancer, the importance of both the choice and the proper definition of these endpoints has been highlighted by cases in which discrepancies between results from different endpoints in the same trial, or for the same endpoint variously defined in different trials, hindered straightforward interpretation of results [28]. Based on a systematic review of the literature, Punt et al. have recommended that disease-free survival (DFS) be defined as the time from randomization to any of eight possible events (Table 3) [28]. DFS is the most appropriate primary endpoint for adjuvant trials in colon cancer, because it includes all clinically relevant events and provides little opportunity for bias. However, because neoadjuvant therapy continues to be a standard of care in rectal cancer, for this disease, an endpoint that includes locoregional failure at the time of surgery assumes importance. Even though this endpoint has often been named DFS [31,32], arguably it is more akin to event-free survival, in whose definition must be included as an event of interest disease progression that precludes surgery (with randomization performed before neoadjuvant therapy) [33]. A recent thorough discussion on endpoints in rectal cancer is available for the interested reader [34]. 

### 3.2. Validation of the Endpoint as a Surrogate

Surrogate endpoints are metrics that can replace previously accepted primary endpoints of interest, such as OS. Colon cancer has spearheaded the search for validated surrogate endpoints that could replace OS in phase 3 trials [33,35,36]. Although different methods have been proposed to validate surrogate endpoints, the currently preferred approach is based on a meta-analysis using individual patient data (IPD) from randomized trials [37]. This approach allows checking the plausibility of randomization sequences, verifying data integrity and consistency, fitting statistical models (copulas) of the correlation between the surrogate and the true endpoint, adjusting analyses for baseline prognostic covariates, performing subset analyses and performing sensitivity analyses using alternative definitions of endpoints. Essentially, IPD surrogacy analyses assess whether (1) a candidate surrogate (e.g., DFS) is associated with the final endpoint (e.g., OS) in individual patients, and (2) the treatment effect on the candidate surrogate can be used to reliably predict the treatment effect on the final endpoint [38]. For DFS and OS, the treatment effect is represented by the hazard ratio for each of these time-to-event endpoints. Condition (1), also known as “patient-level surrogacy”, requires that DFS be a prognostic factor for OS after adjusting for confounders, and this can easily be tested with IPD from any series, including non-randomized patient cohorts. The strength of patient-level surrogacy is often quantified by a measure of correlation, such as Spearman’s rank correlation coefficient. In the absence of IPD, studies based on published data have to rely on medians as a proxy for results in individual patients. Condition (2), also named “trial-level surrogacy”, requires strong levels of association between hazard ratios for DFS and OS. The strength of trial-level surrogacy is quantified by different variants of R^2^, the coefficient of determination [37,39]. Values of correlation coefficients and R^2^ range from 0 to 1, with higher values indicating stronger associations and being required for surrogate validation. Another useful metric obtained more reliably in IPD analyses is the “surrogate threshold effect”, defined as the minimum treatment effect on the surrogate expected to lead to a statistically significant treatment effect on the final endpoint. 

The seminal work by the Adjuvant Colon Cancer Endpoints (ACCENT) Group, led by the late Daniel Sargent, established DFS—with a minimum of 3 years of median follow-up—as a validated surrogate for OS in stage III colon cancer [33,40]. Results for stage II colon cancer have been less convincing, with levels of association between DFS and OS (and between their hazard ratios) that are lower than desirable [33,40]. To our knowledge, no surrogate endpoint has been formally validated for rectal-cancer trials [34]. It should be noted that the definition of DFS used by the ACCENT group differs slightly from the one recommended by Punt et al. [28] (and is more akin to relapse-free survival [28] and the one mentioned in the Food and Drug Administration (FDA) guidance on endpoints [33]). The IDEA collaboration used yet another definition of DFS, excluding only a second primary other than colorectal cancer [3]. Despite these conceptual differences, the practical impact of these definitions in the analysis of DFS or any other time-to-event endpoint is not known. Arguably, these have a trivial impact when considered in aggregate, for example, when a meta-analysis is conducted.

Table 4 displays advantages and disadvantages of endpoints commonly used in oncology, whether in the adjuvant or advanced setting. Since we refer to general properties of these endpoints, Table 4 displays their advantages and disadvantages in general, regardless of whether they are used as primary or secondary, surrogate or final endpoints in a given trial. Given the limitations of time-to-event endpoints, there is tremendous interest in introducing novel endpoints for drug development., especially biomarkers. Surrogate biomarkers can arguably expedite drug development by providing a more sensitive and efficient method to detect recurrences, thus improving statistical power. It should be noted that potential surrogate biomarkers must be prognostic biomarkers that are associated with OS or other time-to-event endpoints at the patient and, and their validation entails demonstration of such an association also at the trial level, as explained above. Noteworthy among potential surrogate biomarkers in the adjuvant setting are circulating tumor cells (CTCs) and DNA. There is considerable evidence that CTCs and circulating tumor DNA are prognostic for recurrence in early or locally-advanced colorectal cancer, since lower levels of these biomarkers at baseline and decreasing levels after treatment are generally associated with better outcomes [41,42,43]. On the other hand, there is no current evidence that these biomarkers serve as validated surrogates in this setting, and further research is needed before they can be reliably implemented as surrogate endpoints. Likewise, other potential surrogate biomarkers are under investigation in colon and rectal cancers, including pathological and image-determined complete responses, platelet-to-lymphocyte and neutrophil-to-lymphocyte ratios and inflammatory and immunological markers. We encourage clinical trialists to add assessment of these biomarkers in a prospective fashion, thus paving the way for future validation studies.

### 3.3. Special Note on “Adjuvant” Trials for Resection of Liver Metastasis 

The evidence on the use of systemic therapy perioperatively or after curative-intent surgery for resectable colorectal liver metastasis has been discussed [44]. In these settings, it is important to further consider when randomization is performed, and the assessment of endpoints started. Trials assessing systemic therapy after resection of liver metastases present fewer difficulties, because randomization can—and ideally should—take place after surgery, thus ensuring adequate balance in prognostic factors in general and those related to the surgical procedure. However, the need remains to consider the start date for assessing endpoints. For example, in some cases, the date used to define DFS is different from the date of randomization (with this difference being as large as a median of 19.5 days [45]); ideally, DFS should be defined from the date of randomization. Moreover, in some cases, there is disease progression between resection and the beginning of systemic therapy, a further reason to use randomization date as the start date for DFS [46,47]. In trials of perioperative therapy, randomization usually takes place before surgery, and the primary endpoint is often called progression-free survival [48,49]. Additional issues that may require attention in such trials include quality control related to surgery [50], potential differences in assessment schedules for different treatment strategies (e.g., upfront surgery versus surgery after preoperative therapy) [44], the degree of subjectivity when defining resectability and potential differences between preoperative and intraoperative assessment of resectability [51]. Additional, specific considerations may be needed in trials of local therapy, such as intra-arterial chemotherapy, selective internal radiotherapy and radiofrequency ablation, as well as in trials of initially unresectable metastatic disease.

### 3.4. Patient Follow-Up

Adequate definition and validation of endpoints are necessary steps in drug development, but other points merit consideration, particularly in adjuvant therapy. One of these is prolonged follow-up of the patients to ensure that the events of interest are captured reliably and without undue delay. The ACCENT Group authors have cautioned that a follow-up of 6 to 8 years is important to confirm an OS benefit in adjuvant colon-cancer trials, especially in light of changing treatment standards, stage migration [52] and the prolongation of post-recurrence survival [53]. To this end, it is crucially important that patients be followed up reasonably frequently and consistently in all randomized arms of comparative trials, that methods of assessment be sensitive and unbiased, and that “dropouts” be minimized. Many trials, unfortunately, discontinue follow-up after a given time, thus precluding the possibility of collecting precious information on long-term events, and the impact of treatment on these events. 

## 4. Optimizing Aspects Related to Implementation of Clinical Trials

### 4.1. Optimizing Speed

One of the key issues in current drug development is speed. With progressively increased understanding of cancer biology and the number of potentially useful anticancer agents, there is mounting competition among stakeholders in this field, as well as increasing pressure on investigators and institutions, which need to be agile and at the same time comply with norms and regulations. Adjuvant therapy is typically a late addition to the portfolio of a given anticancer agent, given the need to first ensure its efficacy in the advanced disease setting. The recent pandemic of the novel coronavirus, SARS-CoV-2, has shown us that clinical trials can be conceived and deployed in a more expedited manner than has usually been the case in oncology and other areas [54]. Among the many roadblocks to the optimization of trial design and conduct with the aim of gaining speed are the often exaggerated or undue concerns with bureaucracy, data quality, data missingness, authorship and data “ownership”. Some of these concerns are addressed below, and the challenge facing investigators is how to counterbalance overcoming these concerns and maintaining methodological rigor and scientific integrity. 

### 4.2. Optimizing Data Quality

One might easily think that optimizing data quality entails collecting more data and making sure they are completely error-free. In fact, it is difficult to agree on what constitutes optimal data quality; in the absence of an objective definition, we propose that in relation to data collected for a clinical trial this means “not too little, not too much, certainly not biased, possibly with some random error”. The prohibitively increasing costs of clinical trials have been a matter of concern for some time [55]. Even though the contribution of clinical trials to the overall cost of drug development is not known with certainty, recent estimates suggest that pivotal trials leading to FDA approval have a median cost of US$19 million, and such costs are even higher in oncology and cardiovascular diseases, as well as in trials with long-term clinical outcomes, such as OS [56]. The extent to which these skyrocketing costs per patient depend on individual components of clinical-trial conduct can vary substantially, and likely so when pivotal trials are compared with investigator-led trials. In the former, considerable resources are spent in making sure that the collected data are error-free, even though random errors play nearly no role in creating biased results in sufficiently large randomized trials [57]. Site visits, usually including source-data verification and other types of quality-assurance procedures—alongside centralized monitoring through data management—make intuitive sense, but their cost has become exorbitant in the large multicenter trials that are typically required for the approval of new agents [58]. It has been estimated that for large, global clinical trials, leaving aside site payments, the cost of on-site monitoring represents about 60% of the total trial [59]. The most time-consuming and least efficient activity is source-data verification, which can take up to 50% of the time spent for on-site visits. A large retrospective study of 1168 industry-sponsored clinical trials has shown that only 1.1% of all data were changed as a result of source-data verification [60]. These efforts would be justified if monitoring activities had a demonstrated impact on patient safety or on trial results [61], but there is scant evidence for that, and randomized studies assessing more intensive versus less intensive monitoring have not shown significant differences in terms of clinically relevant treatment outcomes [62,63,64]. 

In contrast, systematic errors (those that introduce bias in the comparison between treatment groups) can have a huge impact on the trial results [57]. These errors have to be prevented by design, or else they must be detected and corrected. We and others, including regulatory agencies, advocate the use of risk-based monitoring and central statistical monitoring for the detection of systematic errors and fraud in clinical trials [65,66]. Central statistical monitoring has the potential to decrease costs and dramatically increase the efficiency of clinical trials, and the reader is referred to reviews on this subject [67,68].

### 4.3. Optimizing Collaboration

One way to advance clinical research is to foster collaboration, or at least strike an efficient balance between collaboration and healthy competition, among researchers and institutions with a common interest. We do not wish to criticize the current clinical research environment with regard to its publication requirements or, more broadly, academic expectations, but rather focus on the need to collaborate in order to promote scientific advances that translate into better patient care. This spirit of collaboration is famously epitomized by pediatric oncology, cooperative groups and some countries and world regions with the suitable infrastructure and philosophy for its flourishing [69], but is often absent in medical oncology at large, particularly in the setting of sponsored drug development. Fortunately, adjuvant treatment of colon cancer offers an emblematic recent example of prospective international collaboration toward defining the ideal duration of oxaliplatin-based chemotherapy [3]. Nevertheless, several obstacles can be identified that hinder effective international collaboration, including scientific, financial and logistic ones [70]. Of note, the spirit of collaboration needs to go beyond inter-institutional boundaries, because in practice there are several opportunities for intra-institutional barriers, for example, in the dialogue between clinical researchers and statistical departments. The history of oncology suggests that statisticians, epidemiologists and other quantitative methodologists with specialized medical-subject-matter knowledge are an integral part of a successful research team, and that cross-pollination of knowledge among departments is key to success. Finally, data sharing remains suboptimal in medicine, despite several recent efforts to increase this ethically imperative practice [71].

### 4.4. Optimizing Patient-Centricity

In our opinion, a final key component going forward is the involvement of patients (current and future, i.e., society at large) in trial design and interpretation, as well as in the drug-approval and reimbursement processes. This goes beyond patient advocacy, notwithstanding the importance of this field of activity. Since “patients are in a unique position to describe the outcomes that matter to them, to challenge presumptions about their health aspirations and to inform regulatory processes about the potential positive or negative effects of new and existing health technologies” [72], their increasing contribution to drug development is particularly valuable. There are many potential ways to involve patients in different steps of drug development [72,73,74,75,76], and here we focus on only one of these: their expectations and goals when choosing among competing therapies. If known, these same expectations and goals can be used to design new trials.

Conventional statistical methods used in clinical trials take into account the primary endpoint and each secondary endpoint separately. Not infrequently, there are discrepant results for different endpoints, even though major decisions and drug approval is often based on the primary endpoint. Moreover, clinical trials often use composite endpoints that integrate different types of events in the assessment (see Table 3). A composite endpoint takes into account the event first occurring in a given patient, thus ignoring subsequent events. Yet, a locoregional recurrence is far less serious than a distant metastasis (let alone death), but a treatment may be found superior to another in terms of disease-free survival because it delays local recurrence without having any effect (or even possibly a deleterious effect) on the time to distant metastasis or death. A second problem with the current statistical framework is that the association between results for different endpoints is not taken into account. Nevertheless, understanding the association between different outcomes, especially the relationship between efficacy and toxicity, is needed for a meaningful assessment of competing interventions. There may be, for example, independence between efficacy and toxicity; in breast cancer, anthracyclines induce cardiac toxicity that is more likely to occur in patients who are elderly or who have cardiac risk factors, but apparently independently of whether these drugs also produce an antitumor effect in these patients. In some cases, there is a positive association between response and toxicity; for instance, epidermal growth factor receptor inhibitors induce severe skin rash that is associated with response [77]. In other cases, the association is negative; for instance, patients with enzyme deficiencies may experience excessive toxicity that leads to stopping fluoropyrimidine- and irinotecan-based therapy, thus being unlikely to derive benefit from treatment [78,79]. Finally, a third problem with the current framework is that individual priorities with regard to different outcomes from treatment cannot be reliably taken into account just by looking at “average” results for different endpoints in isolation and without due consideration to a hierarchy of individual priorities for a given person.

Until recently, we lacked a formal statistical framework that allowed joint assessment of different endpoints. The development of a statistical framework that permits rigorous joint analysis of several endpoints would allow physicians, patients and other stakeholders to make decisions based on the totality of the available information from the trial. More specifically, this joint assessment could allow three important advances in the design and analysis of randomized trials: (1) taking into account different events and endpoints under a single statistic, (2) taking into account the association between different outcomes and (3) prioritizing outcomes in a patient-centric hierarchical fashion. We have proposed a method, named “generalized pairwise comparisons” (GPC), which can address these issues [80]. By comparing all pairs of patients that can possibly be formed by taking one patient from the experimental and one from the control arm of a randomized trial, GPC leads to the computation of a statistic called “net benefit”, which is the net chance of a better outcome with the experimental treatment than with the control treatment, given a certain hierarchy of priorities among outcomes. This hierarchy of priorities entails choosing the endpoints of interest in the order of priority for a given person, as well as indicating thresholds of clinical significance. For example, one may indicate interest in “any gain in survival” (i.e., a threshold of 0), “only in a survival gain longer than 6 months” (a threshold of 6 months), or any other threshold.

We will illustrate the method with an example coming from metastatic pancreatic cancer [81], but work is ongoing to apply GPC to one of the trials included in the IDEA Collaboration [82]. In the *Partenariat de Recherche en Oncologie Digestive* (PRODIGE) 4 trial, 342 patients were randomly assigned to receive the combination of oxaliplatin, irinotecan, fluorouracil and leucovorin (FOLFIRINOX) or gemcitabine as first-line therapy for metastatic pancreatic cancer [81]. The primary endpoint was OS, and the trial found a hazard ratio of 0.57 (*p* < 0.001) in favor of FOLFIRINOX. However, the combination was more toxic than single-agent gemcitabine, since the frequency of grade ≥ 3 treatment-related adverse events was 69% with FOLFIRINOX and 60% with gemcitabine, even though this difference did not reach statistical significance (*p* = 0.083). Therefore, the question may arise, for example, of whether the OS benefit is offset by the increased toxicity. Answering this question with the help of GPC entails specifying which of these two outcomes is more relevant to a given individual. Moreover, a threshold of clinical significance can be used for OS. In an analysis considering OS as the first priority and with a threshold of 2 months, and toxicity as the second priority, the probability of a random patient having a better outcome with FOLFIRINOX was 59.2%, whereas the corresponding probability was 34.4% with gemcitabine; thus, the net benefit was 59.2%–34.4% = 24.7% (*p* < 0.001) [83]. This analysis ignores additional outcomes, but they can also be accounted for when using GPC. This method is particularly relevant for informing treatment decisions at the individual level, because some patients with incurable disease will prioritize survival, whereas others will prefer to avoid toxicity or prioritize quality of life.

## 5. Conclusions

We can reasonably expect that several novel agents with activity in the (neo)adjuvant therapy of colon and rectal cancer will become available in the near future. Designing the best possible clinical trials of these agents will require the use of the most efficient and reliable trial designs, biomarkers and endpoints that may capture treatment benefit and lead to successful use of novel agents in clinical practice. In this sense, colorectal cancer does not differ appreciably from other disease areas in medical oncology or, for that matter, even from other specialties. On the other hand, colorectal cancer is one of the prime examples in oncology about the worth of (neo)adjuvant therapy, the need to have evidence from well-designed, conducted and analyzed trials, and the need for collaboration across institutions and even countries. We urge clinical trialists, whether physicians, statisticians or any other professional background, to collaborate, involve patients and strike a healthy balance between time-tested and novel methods, as both will be needed going forward.

## Figures and Tables

**Figure 1 cancers-12-03442-f001:**
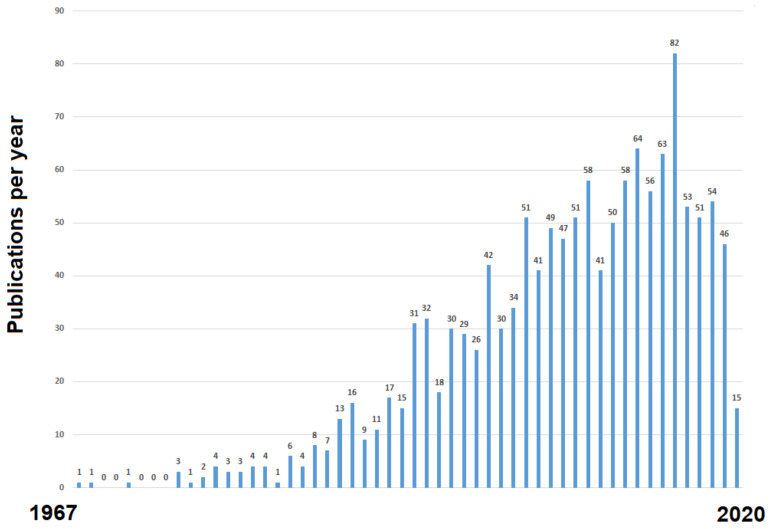
Evolution of publications per year as shown by a PubMed search on 29 October 2020 for the terms colorectal neoplasms [MESH] AND drug therapy [MESH] AND “adjuvant” and the filter “clinical trials”. [MESH] refers to the search tag used to indicate “medical subject heading”.

**Figure 2 cancers-12-03442-f002:**
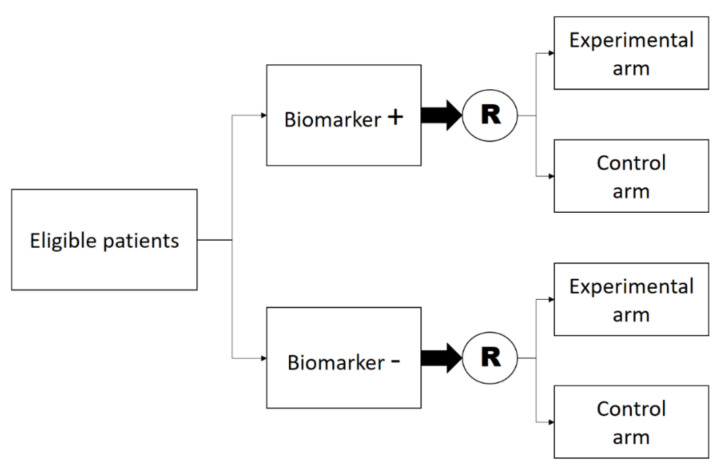
Generic design of a biomarker-stratified trial. “Biomarker +” and “Biomarker −“ refer to patients with presence and absence of biomarker respectively, and “R” denotes randomization.

**Table 1 cancers-12-03442-t001:** Key features of explanatory and pragmatic clinical trials.

Feature	Explanatory Trials	Pragmatic Trials
Key assessment	Efficacy	Effectiveness
Main sponsor	Pharmaceutical companies	Academic groups
Primary purpose	Regulatory approval	Clinical practice
Patient profile	Fittest patients	All patients
Effect of interest	“Ideal” treatment effect	Actual treatment effect
Endpoint ascertainment	Centrally reviewed	Per local investigator
Preferred control group	Standard of care or placebo (when feasible)	Competing available intervention
Experimental conditions	Strictly controlled	As close as possible to clinical practice
Data collection	Extensive	Key data
Data quality control	Extensive and on-site	Limited or only centralized

**Table 2 cancers-12-03442-t002:** Example of minimization in a trial for which the next patient to be entered has the features shown in the first column, whereas the remaining columns show the numbers of patients with each of the features already randomized to each arm. “Total” does not refer to total number of patients in each arm, but rather to the imbalance explained in the text.

Patient Features for Each Factor	Treatment Policy	
	Stop	Continue
Chemotherapy = FOLFOX	55	54
T stage = T4	22	23
N stage = N2	19	16
Center = Jules Bordet Institute	4	3
Total	100	96

**Table 3 cancers-12-03442-t003:** Definition of selected time-to-event endpoints for adjuvant therapy [28].

Events of Interest in the Analysis of Each Endpoint	Endpoint		
	Disease-Free Survival	Recurrence (or Relapse)-Free Survival	Time to Recurrence
Locoregional recurrence	X	X	X
Distant metastasis	X	X	X
Second primary cancer of the same site	X		
Other second primary cancer	X		
Death from cancer of the same site	X	X	X
Death from other cancer	X	X	
Non-cancer death	X	X	
Treatment-related death	X	X	

**Table 4 cancers-12-03442-t004:** Conceptual representation of the advantages and disadvantages of endpoints commonly used in oncology.

Endpoints	Ease of Measurement	Time of Measurement	Potential for Bias	Statistical Power	Clinical Relevance
Overall survival	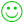	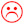	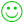	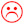	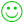
Quality of life	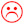	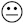	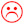	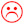	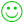
TTP/PFS/DFS	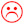	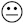	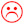	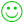	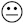
Response rate	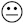	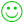	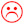	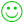	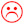
Biomarker	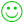	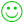	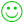	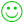	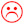

DFS, disease-free survival; PFS, progression-free survival; TTP, time to progression. 
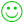
, 
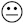
 and 
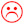
 indicate that corresponding endpoints perform well, fairly or poorly respectively, with regard to characteristics indicated as column headings.

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
