# Peer review of "Statistical Considerations for Trials in Adjuvant Treatment of Colorectal Cancer"

_cancers, 2020, doi:10.3390/cancers12113442_

Round 1
Reviewer 1 Report
This manuscript provides a nice summary of important concepts that should be taken into account when designing a clinical trial. It provides a good overview of classical concepts and guides the reader through some innovations by providing examples with references. I congratulate the authors with this well-written work and I fully agree with all points they make.
The article is good and acceptable in its current form. I have no comments, only some suggestions that the authors might wish to consider:
- Perhaps it could be noted that randomization makes unknown factors comparable across treatment groups only if the sample size is large enough.
- It would be good to mention the possibility of adding a random element to the minimization procedure, so that the allocation sequence will be less predictable.
Some minor/typographical remarks:
- line 53 "investigators attempts": should be investigators attempt
- line 58 "epitomize": perhaps a less complicated wording would make the article better accessible to an international (non-native English speaking) audience.
- line 267 "DFS be an independent prognostic factor for OS": I personally find the word `independent' here a bit confusing. I would like to suggest to make this sentence a bit more explicit. So for example "DFS be prognostic for OS after correction for confounders" or so, or just leave out the word `independent'.
- line 355 "surmise": again this could be made easier to read by saying something like "our assessment is that".
- line 431 "In same cases": should be "In some cases"
Author Response
We thank the reviewer for the positive appreciation of the manuscript and the useful suggestions to improve it.
With regard to a large sample size as a prerequisite for randomization to balance unknown prognostic factors across treatment groups, we have added such a note on page 5, line 162.
With regard to the possibility of adding a random element to the minimization procedure, this is now mentioned on page 6, line 195.
We have corrected the typographical errors identified by the reviewer:
- line 53 "investigators attempts": should be investigators attempt
- line 58 "epitomize": perhaps a less complicated wording would make the article better accessible to an international (non-native English speaking) audience.
- line 267 (now 292) "DFS be an independent prognostic factor for OS": I personally find the word `independent' here a bit confusing. I would like to suggest to make this sentence a bit more explicit. So for example "DFS be prognostic for OS after correction for confounders" or so, or just leave out the word `independent'.
- line 355 (now 391) "surmise": again this could be made easier to read by saying something like "our assessment is that".
- line 431 (now 467) "In same cases": should be "In some cases"
Reviewer 2 Report
Overall:
The authors give a readable and up-to-date overview of various aspects that come into play when conducting an adjuvant trial in colorectal cancer, or planning to do so.
It is not entirely clear what kind of article this is. It is not a traditional scientific article as there are no new results presented here. Rather they give an overview of relevant existing knowledge. However it is also not a survey in the traditional sense, because that would require much more exhausting citing of literature.
All in all I think the best way to view the article is as a ‘how-to-guide’ helping local investigators who are thinking about conducting a randomized trial comparing two adjuvant treatment options in colorectal cancer.
As such, the article certainly has value. There are many aspects to think of when doing a trial and it is certainly helpful to have all (or many) of them listed in one place, which is what the current article. Also the points in the article are scientifically sound and reflect current consensus as far as I can tell
However I think it would be helpful for the potential reader if the authors make it more clear from the beginning (in the introduction and/or abstract) what is the purpose of the article and who is the intended target audience.
Specific points:
Line 290: “Given the limitations of time-to-event endpoints, there is tremendous interest in developing novel endpoints for drug development, especially biomarkers (Figure 3).”, and paragraph following it:
- It may make sense to move this line up a bit so that it appears earlier in section 3.2; now it is a bit unclear why we spend so much time on validating surrogates until we hit this line
- There are two examples now: circulating tumor cells and DNA. In the formulation it is not clear if ‘DNA’ is just ‘DNA’ or ‘circulating tumor DNA’. In both cases the authors could elaborate on how they are used as endpoint. Is it e.g. a decrease in CTCs that is prognostic of better outcome?
- Having more examples of alternative endpoints that are currently investigated would not hurt
- Advise on line 300-302 is very good, keep this in!
- Here we are talking about biomarkers used as endpoints. Earlier in the article there was talk about use of biomarkers used as stratification factors. It would maybe help the reader if the distinction is perhaps made a bit clearer. Also there are situations where the two uses overlap. For instance: when using decrease or increase in CTCs as surrogate endpoint one obviously needs a baseline measurement and this in turn could also be used as a stratification factor. The authors could add a short paragraph discussing pros and cons of doing this.
- Figure 3 mentions PFS while this was not defined in Table 3, which talks about RFS. Please reconcile this, preferably by extending Table 3 with a definition of PFS (and other relevant endpoints if there are any).
Line 311-315: ‘For example, in some cases the date used to define DFS is different from the date of surgical resection (with this difference being as large as a median of 31319.5 days [45]); in others, DFS is defined from the date of randomization, which is more appropriate as a general rule, as in some cases there is disease progression between resection and the beginning of systemic therapy [46,47]’
I find the whole section (‘3.3. Special note on “adjuvant” trials for resection of liver metastasis’) a bit confusing. The main point seems to be that one should think about where to start counting for time to event endpoints, but no clear answer emerges. Also maybe that is not the main point? Please clarify this section. The sentence quoted above already contains several reasons for my confusion that I will list separately:
- The use of ‘in others’ suggests a contrast. The ‘other’ trials from the second half of the sentence start counting at date of randomization suggesting that the trials form the first half of the sentence use some other date. But what is it? One would expect the date of surgery, which is the other ‘natural’ choice besides date of randomization but apparently that is not the case here given that the first half of the sentence says that the start date of DFS and the date of surgery are often different. But this would also be the case in trials that used the date of randomization as a start date, as in the second half.
- Defining DFS from date of randomization is more appropriate as a general rule. I agree, but I am not sure if I follow the argument (progression between resection and the beginning of systemic therapy). How are these progressions relevant? Is the idea that these might occur before randomization?? In that case please say so. Since randomization both before and after surgery is discussed this is not clear.
- Also: when discussing what is ‘more appropriate’ it is good to know what are the options we are considering. I can think of using date of randomization as a start date and using date of surgery as a start date, but (see also the first bullet point) the current formulation suggests that some trials use a yet different third starting point. What is this?
- Also the other ‘standard’ argument for starting to count at randomization (‘that is the last moment at which the patients in both arm are still ‘the same’ (hence comparable)’) is not mentioned
- The use of ‘as a general rule’ suggests that while it starting to count at randomization is better in most situations, there are specific situations where starting to count somewhere else, e.g. at surgery is better. Please list these situations and the argument.
Line 399-402: “The history of oncology has shown that statisticians, epidemiologists and other quantitative methodologists with specialized medical-subject-matter knowledge are an integral part of a successful research team, and that cross-pollination of knowledge among departments is key to success.”
I certainly agree and feel flattered, but it would be good to have some reference to back up this claim, if it can be found.
Line 411-412: “There are many potential ways to involve patients in different steps of drug development, and here we focus on only of these”
This is like my previous point but more serious: it is ok to focus only of one of these but please try and find some references to other ‘potential ways’ and include them. Many researchers would like to participate patients more but don’t know how and on the other hand there is literature on exactly that that has trouble finding its way to readers. Your article could be the perfect bridge between the two.
Line 431: ‘In same cases’: this must be in some cases.
Line 267: ‘independent prognostic factor’.
Independent of what? I recommend not usising this expression (either omit the independent or replace it with a more precise/explicit alternative). I believe the other reviewer also said something about this although I don’t know what.
Author Response
We thank the reviewer for the overall positive appraisal of our manuscript. Perhaps due to failure on our part to indicate this in the submission process, this is an invited work on a series of papers dealing with the adjuvant therapy of colorectal cancer, our piece being the one dedicated to statistical aspects. Given this context, we do not believe an extensive explanation is required, but we agree with the view that the article is as a ‘how-to-guide’ helping investigators who are planning a randomized trial of adjuvant therapy for colorectal cancer, and this is now indicated in the Introduction. Other changes are shown below in red.
Specific points:
Line 290: “Given the limitations of time-to-event endpoints, there is tremendous interest in developing novel endpoints for drug development, especially biomarkers (Figure 3).”, and paragraph following it:
- It may make sense to move this line up a bit so that it appears earlier in section 3.2; now it is a bit unclear why we spend so much time on validating surrogates until we hit this line. Please see below for changes that were made in order to address the several issues identified in relation to this section.
- There are two examples now: circulating tumor cells and DNA. In the formulation it is not clear if ‘DNA’ is just ‘DNA’ or ‘circulating tumor DNA’. In both cases the authors could elaborate on howthey are used as endpoint. Is it e.g. a decrease in CTCs that is prognostic of better outcome? We have added some explanations.
- Having more examples of alternative endpoints that are currently investigated would not hurt We have added some examples.
- Advise on line 300-302 is very good, keep this in!
- Here we are talking about biomarkers used as endpoints. Earlier in the article there was talk about use of biomarkers used as stratification factors. It would maybe help the reader if the distinction is perhaps made a bit clearer. Also there are situations where the two uses overlap. For instance: when using decrease or increase in CTCs as surrogate endpoint one obviously needs a baseline measurement and this in turn could also be used as a stratification factor. The authors could add a short paragraph discussing pros and cons of doing this. We have made several modifications throughout the manuscript aiming at better explaining what is meant by a "biomarker" in different context. For example, a short paragraph on biomarker definitions is now provided in section 2.3, and other changes are made elsewhere.
- Figure 3 mentions PFS while this was not defined in Table 3, which talks about RFS. Please reconcile this, preferably by extending Table 3 with a definition of PFS (and other relevant endpoints if there are any). We have tried to clarify that Table 3 deals exclusively with endpoints for adjuvant therapy (where PFS plays no role), whereas Figure 3 extends to advanced disease.
Line 311-315: ‘For example, in some cases the date used to define DFS is different from the date of surgical resection (with this difference being as large as a median of 19.5 days [45]); in others, DFS is defined from the date of randomization, which is more appropriate as a general rule, as in some cases there is disease progression between resection and the beginning of systemic therapy [46,47]’
I find the whole section (‘3.3. Special note on “adjuvant” trials for resection of liver metastasis’) a bit confusing. The main point seems to be that one should think about where to start counting for time to event endpoints, but no clear answer emerges. Also maybe that is not the main point? Please clarify this section. The sentence quoted above already contains several reasons for my confusion that I will list separately:
- The use of ‘in others’ suggests a contrast. The ‘other’ trials from the second half of the sentence start counting at date of randomization suggesting that the trials form the first half of the sentence use some other date. But what is it? One would expect the date of surgery, which is the other ‘natural’ choice besides date of randomization but apparently that is not the case here given that the first half of the sentence says that the start date of DFS and the date of surgery are often different. But this would also be the case in trials that used the date of randomization as a start date, as in the second half. We agree with the reviewer that this passage was confusing and have attempted to improve it.
- Defining DFS from date of randomization is more appropriate as a general rule. I agree, but I am not sure if I follow the argument (progression between resection and the beginning of systemic therapy). How are these progressions relevant? Is the idea that these might occur beforerandomization?? In that case please say so. Since randomization both before and after surgery is discussed this is not clear. We hope to have clarified this issue with the changes made.
- Also: when discussing what is ‘more appropriate’ it is good to know what are the options we are considering. I can think of using date of randomization as a start date and using date of surgery as a start date, but (see also the first bullet point) the current formulation suggests that some trials use a yet different third starting point. What is this? We hope to have clarified this issue with the changes made.
- Also the other ‘standard’ argument for starting to count at randomization (‘that is the last moment at which the patients in both arm are still ‘the same’ (hence comparable)’) is not mentioned. We believe the sentence "thus ensuring adequate balance in prognostic factors in general and those related to the surgical procedure" is analogous to the reviewer's suggestion.
- The use of ‘as a general rule’ suggests that while it starting to count at randomization is better in most situations, there are specific situations where starting to count somewhere else, e.g. at surgery is better. Please list these situations and the argument. We hope to have clarified this issue with the changes made.
Line 399-402: “The history of oncology has shown that statisticians, epidemiologists and other quantitative methodologists with specialized medical-subject-matter knowledge are an integral part of a successful research team, and that cross-pollination of knowledge among departments is key to success.”
I certainly agree and feel flattered, but it would be good to have some reference to back up this claim, if it can be found. We appreciate this candid comment, but unfortunately have not found a good reference encompassing the various aspects of our statement (although Gehan's piece in Biometrics 1980;36(4):699-706 and Califf's in Clin Trials 2016;13(5):471-7 get close to it). For this reason, we have replaced "has shown" by "suggests".
Line 411-412: “There are many potential ways to involve patients in different steps of drug development, and here we focus on only of these”
This is like my previous point but more serious: it is ok to focus only of one of these but please try and find some references to other ‘potential ways’ and include them. Many researchers would like to participate patients more but don’t know how and on the other hand there is literature on exactly that that has trouble finding its way to readers. Your article could be the perfect bridge between the two. Once again, we thank the reviewer for what appears to be a heart-felt suggestion. Given the vast literature on this subject, we have added only a few references in the hope they will be useful.
Line 431: ‘In same cases’: this must be in some cases. We have corrected the error.
Line 267: ‘independent prognostic factor’.
Independent of what? I recommend not usising this expression (either omit the independent or replace it with a more precise/explicit alternative). I believe the other reviewer also said something about this although I don’t know what. We have addressed this, also following the concern from the other reviewer.
Reviewer 3 Report
Review of a significant number of publications (79), however, a large portion (19) of those are co-authored by one of the authors (Buyse) of this manuscript.
The following comments are suggested for consideration during revisions.
Title is very long and confusing
Authors didn’t explain well what is the existing problem that requires optimization of statistical aspects? Why is the concentration on the adjuvant treatment? Is the problem specific to adjuvant treatment only? If so, describe. Is the optimization relevant to colorectal cancer only – if yes, then state so. Is this optimization not applicable to other treatments and other types of cancer? Not clear.
Keywords don’t reflect optimization of statistical aspect (re-visit keywords).
Authors are stating: “ Research output on adjuvant therapy for colorectal cancer seems to have stalled in the last few years, as suggested by a simple search for published articles (Figure 1)”. It is not clear how research output on adjuvant therapy stalled based on the number of publications. Numbers of publications are not even given in Figure 1. Do we measure research output only by the number of publications? Other types of evidence are needed to confirm that research is stalled indeed. Explain [MESH].
Figure 2 – explain in footnotes what ‘Biomarker+ ‘and ‘Biomarker- ‘ mean.
Figure 3 needs to be re-labeled as Table 3. What is the meaning of the faces in this Figure/Table? Explain in footnotes. Correct the headings of the columns.
Table 3 – hard to read and understand column ‘events of interest’. Table 3 needs to be modified/re-done significantly.
Introduction is extremely short, no background information is given on what is the need for this publication?
Line 53 page 2 – ‘investigators attempt’
Table 1 needs to be explained a little more in the text
Table 2 needs better explanation in the text. It also needs a better title. ’Example of minimization’ – explain. Table caption should be self-explanatory, don’t refer in the caption to the text for details. Explain what is ‘overall imbalance’.
Explain what permuted blocks are.
Table2 is mentioned in the text as Table 3.
Figure 3 –column headings need to be fixed.
Explain why is focus on phase 3 trials?
Explain what is surrogate endpoint?
The overall significance and novelty of this review is not clearly stated.
Author Response
We thank the reviewer for the constructive criticism. Perhaps due to failure on our part to indicate this in the submission process, this is an invited work on a series of papers dealing with the adjuvant therapy of colorectal cancer, our piece being the one dedicated to statistical aspects. We believe this clarification will address some of the concerns raised, including the number of self-citations, the focus on adjuvant treatment, and the overall goal of this work. Other changes are shown below in red.
Title is very long and confusing. We have added a comment proposing a new title.
Authors didn’t explain well what is the existing problem that requires optimization of statistical aspects? Why is the concentration on the adjuvant treatment? Is the problem specific to adjuvant treatment only? If so, describe. Is the optimization relevant to colorectal cancer only – if yes, then state so. Is this optimization not applicable to other treatments and other types of cancer? Not clear. Please see the opening statements above.
Keywords don’t reflect optimization of statistical aspect (re-visit keywords). The initial key words are MESH terms, which are indeed not necessarily adequate. We have added some additional ones (not all MESH terms).
Authors are stating: “ Research output on adjuvant therapy for colorectal cancer seems to have stalled in the last few years, as suggested by a simple search for published articles (Figure 1)”. It is not clear how research output on adjuvant therapy stalled based on the number of publications. Numbers of publications are not even given in Figure 1. Do we measure research output only by the number of publications? Other types of evidence are needed to confirm that research is stalled indeed. Explain [MESH]. We are not sure to understand this suggestion. Our statement is cautious in the use of "seems to have stalled (...) as suggested by". At any rate, we have added "at least" to further indicate our caution. Moreover, Figure 1 shows the number of publications per year, and we believe the number of publications is indeed a good metric of research output. We have now explained the meaning of "[MESH]".
Figure 2 – explain in footnotes what ‘Biomarker+ ‘and ‘Biomarker- ‘ mean. We have added the explanation.
Figure 3 needs to be re-labeled as Table 3. What is the meaning of the faces in this Figure/Table? Explain in footnotes. Correct the headings of the columns. We prefer to leave the choice between Table and Figure to denote this display item and the formatting of headings to the editorial office. We have added an explanation about the meanings of faces.
Table 3 – hard to read and understand column ‘events of interest’. Table 3 needs to be modified/re-done significantly. We have left the original table, with an improved label for the column ‘events of interest’, as well as a proposed new table.
Introduction is extremely short, no background information is given on what is the need for this publication? Please see the opening statements above.
Line 53 page 2 – ‘investigators attempt’ We have corrected the error.
Table 1 needs to be explained a little more in the text. We have added some explanation as suggested.
Table 2 needs better explanation in the text. It also needs a better title. ’Example of minimization’ – explain. Table caption should be self-explanatory, don’t refer in the caption to the text for details. Explain what is ‘overall imbalance’. We have added explanations as suggested.
Explain what permuted blocks are. We have added the explanation as suggested.
Table2 is mentioned in the text as Table 3. We have corrected the error.
Figure 3 –column headings need to be fixed. Please see comment above.
Explain why is focus on phase 3 trials? We believe that clinical trials for adjuvant therapy, with very few exceptions of pilot situations of very complex interventions, need to be randomized, but this is left implicit in the manuscript.
Explain what is surrogate endpoint? We have added the explanation in the opening of section 3.2.
The overall significance and novelty of this review is not clearly stated. Please see the opening statements above.
Round 2
Reviewer 3 Report
Overall, authors did a good job addressing the reviewers' comments. Additional suggestions:
Figure 1:
- This figure still does not include number of publications corresponding to each bar (in original figure in Pubmed each bar is clickable for number of publications). Please make the figure image larger and show numbers of publications on top of each bar in your manuscript. Currently it is impossible to figure out how many publications are being produced on this topic per year.
- Please re-run the Pubmed search to include the information through October 2020. We are 5 months past the last search shown in Figure 1.
- Change to: Evolution of number of publications per year as shown by a PubMed search on 14 June October 26 2020 for the terms colorectal neoplasms [MESH] AND drug therapy [MESH] AND "adjuvant" and the filter “clinical trials”.
4. Introduction, line 25-27:
‘research output on adjuvant therapy for colorectal cancer seems to have stalled in the last few years, at least as suggested by a simple search for published articles (Figure 1). – I still have problem with this statement, please re-phrase it:
‘As suggested by a simple search for published articles (Figure 1), the number of research publications and potentially research output on adjuvant therapy for colorectal cancer has decreased since 2016.'
5. Table 3 – please use only the new version (bottom part in red) of the table. It is much easier to read and interpret the data presented in the table.
6. Figure 3:
a. this figure doesn’t look like a figure to me. I consider this type of data a table. I suggest re-naming it as Table 4.
b. In addition, column headings have not been formatted properly.
c. If color-coding is being used to indicate the level of performance of endpoint, then solid color circles green, red and black) without faces would look more professional.
d. Specify type of endpoints in this figure/table (surrogate, primary, time-to-event)? Not clear which ones are the authors talking about.
7. Page 5, line 177: ‘Pan European Trial Adjuvant Colon Cancer (PETTAC)-3’ – is this a correct abbreviation or should it be ‘PETACC’?
8. CONCLUSION are very general and vague and can be said in this form about any therapy of any cancer. Please add some specific conclusions based on what your review is presenting, especially in terms of specific statistical considerations and specific biomarkers you are describing.
Author Response
Overall, authors did a good job addressing the reviewers' comments. Additional suggestions:
Figure 1:
- This figure still does not include number of publications corresponding to each bar (in original figure in Pubmed each bar is clickable for number of publications). Please make the figure image larger and show numbers of publications on top of each bar in your manuscript. Currently it is impossible to figure out how many publications are being produced on this topic per year. Once again, we thank the reviewer for the suggestions. We now offer two versions of Figure 1. Given the number of years, none is ideal, but both show the number of publications between the years 1967 and 2020.
- Please re-run the Pubmed search to include the information through October 2020. We are 5 months past the last search shown in Figure 1. We have done that for the new figures.
- Change to: Evolution of number of publications per year as shown by a PubMed search on 14 June October 26 2020 for the terms colorectal neoplasms [MESH] AND drug therapy [MESH] AND "adjuvant" and the filter “clinical trials”. We have done that.
4. Introduction, line 25-27:
‘research output on adjuvant therapy for colorectal cancer seems to have stalled in the last few years, at least as suggested by a simple search for published articles (Figure 1). – I still have problem with this statement, please re-phrase it:
‘As suggested by a simple search for published articles (Figure 1), the number of research publications and potentially research output on adjuvant therapy for colorectal cancer has decreased since 2016.' We have rephrased as suggested.
5. Table 3 – please use only the new version (bottom part in red) of the table. It is much easier to read and interpret the data presented in the table. We agree!
6. Figure 3:
a. this figure doesn’t look like a figure to me. I consider this type of data a table. I suggest re-naming it as Table 4. We have changed as suggested.
b. In addition, column headings have not been formatted properly. Font size reduced.
c. If color-coding is being used to indicate the level of performance of endpoint, then solid color circles green, red and black) without faces would look more professional. Unfortunately, we do not have the graphic-design expertise to do that, but would accept further suggestions from the reviewer or editorial office.
d. Specify type of endpoints in this figure/table (surrogate, primary, time-to-event)? Not clear which ones are the authors talking about. Since we refer to general properties of these endpoints, here we refer to them in general, regardless of whether they are used as primary or secondary, surrogate or final. A note has been added to explain this.
7. Page 5, line 177: ‘Pan European Trial Adjuvant Colon Cancer (PETTAC)-3’ – is this a correct abbreviation or should it be ‘PETACC’? Thank you for pointing this out.
8. CONCLUSION are very general and vague and can be said in this form about any therapy of any cancer. Please add some specific conclusions based on what your review is presenting, especially in terms of specific statistical considerations and specific biomarkers you are describing. We agree, and hope the added language can improve the Conclusion, especially considering that this piece is meant to be part of a collection of papers on colorectal cancer.